# Tankyrase (PARP5) Inhibition Induces Bone Loss through Accumulation of Its Substrate SH3BP2

**DOI:** 10.3390/cells8020195

**Published:** 2019-02-22

**Authors:** Tomoyuki Mukai, Shunichi Fujita, Yoshitaka Morita

**Affiliations:** Department of Rheumatology, Kawasaki Medical School, 577 Matsushima, Kurashiki, Okayama 701-0192, Japan; shunic117@gmail.com (S.F.); morita@med.kawasaki-m.ac.jp (Y.M.)

**Keywords:** tankyrase, PARP5, tankyrase inhibitor, SH3BP2, cherubism, osteoclast, osteoblast, osteopenia, PARP1

## Abstract

There is considerable interest in tankyrase because of its potential use in cancer therapy. Tankyrase catalyzes the ADP-ribosylation of a variety of target proteins and regulates various cellular processes. The anti-cancer effects of tankyrase inhibitors are mainly due to their suppression of Wnt signaling and inhibition of telomerase activity, which are mediated by AXIN and TRF1 stabilization, respectively. In this review, we describe the underappreciated effects of another substrate, SH3 domain-binding protein 2 (SH3BP2). Specifically, SH3BP2 is an adaptor protein that regulates intracellular signaling pathways. Additionally, in the human genetic disorder cherubism, the gain-of-function mutations in SH3BP2 enhance osteoclastogenesis. The pharmacological inhibition of tankyrase in mice induces bone loss through the accumulation of SH3BP2 and the subsequent increase in osteoclast formation. These findings reveal the novel functions of tankyrase influencing bone homeostasis, and imply that tankyrase inhibitor treatments in a clinical setting may be associated with adverse effects on bone mass.

## 1. Introduction

### 1.1. Overview of Tankyrase

The poly(ADP-ribose) polymerase (PARP) superfamily catalyzes the ADP-ribosylation of target proteins and regulates various cellular processes, including the maintenance of genome stability, mitosis, and signal transduction [1,2]. Tankyrase-1 and tankyrase-2, which are also called PARP5a and PARP5b, respectively, are members of the PARP family [3,4]. Structurally, both tankyrases comprise five ankyrin (ANK) repeats, a sterile alpha motif, and a catalytic domain [1] (Figure 1). The ANK domain serves as a binding platform for the ADP-ribosylation of substrates. In particular, the consensus hexapeptide motif RxxPDG (where ‘x’ refers to any amino acid) within the ANK domain is necessary for interactions with target proteins [5,6]. Tankyrase-1 and tankyrase-2 are ubiquitously expressed in various tissues. Tankyrases generally have overlapping functions, which have been revealed based on the development of subtle phenotypes following the deletion of either gene [6,7]. Because double knockout mice lacking the genes encoding tankyrase-1 and tankyrase-2 exhibit embryonic lethality [6,8], tankyrase proteins are essential for embryonic development.

Several tankyrase substrates have been identified, including TRF1 [5,9,10], AXIN [5,11], nuclear mitotic apparatus protein (NuMA) [5,10], insulin-responsive amino peptide (IRAP) [5,10], 182-kD tankyrase-binding protein (TAB182) [5,9,10], formin-binding protein 17 (FBP17) [5,12], DNA repair factor MERIT40 [5], SH3 domain-binding protein 2 (SH3BP2) [5,6], CBP80/CBP20-dependent translation initiation factor (CTIF) [13], and peroxiredoxin II (PrxII) [14]. Tankyrases regulate various cellular activities, such as responses to DNA damage, maintenance of telomere length, mitosis, and Wnt- and Notch-mediated signal transduction, by modulating their substrates [1,2,4]. Although there have been substantial advances in tankyrase research, not all of the substrate-mediated pathways are well characterized.

One of the thoroughly studied substrates is AXIN, which is a negative regulator of the canonical Wnt pathway [11]. A previous study revealed that Wnt regulates the nuclear translocation of transcription factor β-catenin and induces the expression of Wnt/β-catenin-responsive genes [11]. In the absence of an activating Wnt signal, GSK3β phosphorylates β-catenin in collaboration with AXIN and APC proteins. The phosphorylated β-catenin is ubiquitinated and then degraded by a proteasome. In this pathway, tankyrase induces the degradation of AXIN and subsequently stabilizes β-catenin, ultimately resulting in the upregulated expression of Wnt/β-catenin-responsive genes [11]. Thus, tankyrase inhibitors function as Wnt/β-catenin inhibitors by stabilizing the negative regulator AXIN [15].

### 1.2. Preclinical Application of Tankyrase Inhibitors in Cancer

Tankyrase has attracted attention as a novel and promising target for cancer treatments [1]. Indeed, various small-molecule tankyrase inhibitors have been synthesized and confirmed to have anti-cancer effects on a wide range of cancers, including colon [16,17], breast [18,19], lung [20], prostate [21], ovarian [22], and liver [23], as well as osteosarcoma [24,25]. Tankyrase inhibitors exert their anti-cancer effects through multiple mechanisms. The inhibition of tankyrase is reported to exhibit an anti-cancer effect against BRCA1/2-deficient and Wnt-dependent cancers as well as a universal anti-cancer effect through the stabilization of TRF1 and a concomitant inhibition of telomerase function [1,2]. The application of tankyrase inhibitors may be particularly useful for treating Wnt-dependent cancers, especially colorectal cancer [16,17]. In Wnt-dependent cancers, Wnt pathways are aberrantly activated by mutational mechanisms, including the inactivation of APC or AXIN, or by activating mutations in β-catenin, all of which lead to constitutive transcription of β-catenin/TCF-regulated genes [11,26,27,28]. In Wnt-dependent cancers, tankyrase inhibitors exert anti-cancer effects by inhibiting the proliferation of cancer cells. Studies that evaluated the effect of tankyrase inhibitors on murine in vivo cancer models are listed in Table 1.

In addition, the mechanism of synthetic lethality should be noted. Cancers with mutation of BRCA1/2 are treatable by PARP inhibitors [29,30,31]. PARPs and BRCA1/2 are both important for DNA repair [32,33]. Neither mutation of BRCA nor inhibition of PARP are lethal because cells can inherently repair DNA damage [32,33]. However, inhibiting both pathways, represented by administration of PARP inhibitors in BRCA1/2-mutated cancers, prevents both DNA repair machineries, resulting in unrepairable DNA damage and subsequent cell death of the mutated cancer cells [32,33]. The PARP1 inhibitor olaparib has been approved and has achieved success in the treatment of BRCA-mutated cancers in the basis of this synthetic lethality approach [29,30,31].

### 1.3. Preclinical Application of Tankyrase Inhibitors in Fibrotic Diseases

Because canonical Wnt signaling is an essential mediator of fibroblast activation and tissue fibrosis [40,41], tankyrase inhibitors represent promising drug candidates for treating fibrotic diseases. The therapeutic effects of tankyrase inhibitor XAV939 and siRNA-mediated knockdown of tankyrases were evaluated in murine models of bleomycin-induced dermal fibrosis and viral TGF-β receptor I-overexpressed fibrosis [42] (Table 2). The inactivation of tankyrases effectively abrogated the activation of canonical Wnt signaling and exerted anti-fibrotic effects [42] (Table 2). Consistent with the in vivo effects on fibrosis, tankyrase inhibitors reportedly also exhibit anti-fibrotic effects in cell culture models of pulmonary fibrosis and keloid formation [43,44]. These findings suggest that tankyrase inhibitors may be applicable to the treatment of a wide range of fibrotic diseases.

### 1.4. Tankyrase-Specific Inhibitors

Tankyrase inhibitors have been classified into two types. Specifically, one type targets the nicotinamide subsite of the tankyrase protein, which is conserved in various PARPs, and the other type targets a unique adenosine subsite that is more potent and specific to tankyrase [46]. Tankyrase inhibitor XAV939 has been categorized as the first type, whereas newly designed drugs (e.g., IWR-1 [47], G007-LK [39,48], and NVP-TNKS656 [49]) have been categorized as the second type. Chemical structures of the inhibitors are presented in Figure 2. Moreover, XAV939, which is widely used as a tankyrase inhibitor in various experimental settings [11], similarly inhibits tankyrase-1, tankyrase-2, PARP1, and PARP2, with IC_50_ values of 95, 5, 74, and 27 nM, respectively, because of the highly conserved nicotinamide subsite among PARP family members [46,47]. Thus, the adenosine subsite may represent a more desirable target for tankyrase-1/2-specific inhibition.

## 2. Effects of Tankyrase Inhibition on Bone

### 2.1. SH3BP2, an Unappreciated Substrate for Tankyrase, Regulates Osteoclastogenesis

SH3BP2 is an adaptor protein that is predominantly expressed in immune cells, including macrophages/osteoclasts and osteoblasts [51,52,53,54]. SH3BP2 functions as a scaffold protein to transduce various intracellular signaling pathways, including those related to protein tyrosine kinases SYK [51,55], PLCγ [56,57], VAV [58], SRC [59], and ABL [60]. SH3BP2 contains an N-terminal pleckstrin homology (PH) domain, a proline-rich domain that binds to Src homology (SH) 3 domain-containing proteins, and a C-terminal SH2 domain that binds to phospho-tyrosine residues [51] (Figure 3a). The *SH3BP2* gene is responsible for the human genetic disease cherubism (OMIM#118400) [61]. Cherubism is an autosomal dominant disease characterized by maxillary and mandibular bone destruction accompanied by increased osteoclast formation [61,62]. Cherubism patients have heterozygous mutations within the peptide sequence RSPPDG that lies between the PH and SH2 domains of SH3BP2. Cherubism-linked mutations to SH3BP2 (e.g., R415G, P418L, P418R, and G420R [61]) prevent tankyrase from recognizing the RxxPDG sequence [5] (Figure 3b,c). These amino acid changes represent gain-of-function mutations because of the resulting dysregulation of substrate recognition by tankyrase [5,6]. Because tankyrase recognizes SH3BP2 via the RxxPDG sequence, the mutated SH3BP2 protein does not undergo tankyrase-mediated ADP-ribosylation or the subsequent E3-ubiquitin ligase RNF146-mediated degradation process [6,63,64] (Figure 3b). Consequently, the mutated SH3BP2 protein accumulates in the cytoplasm, where it aberrantly upregulates SH3BP2-mediated intracellular signaling pathways, such as those related to SYK, PLCγ, VAV, SRC, and ABL [6,51,52,55,56,57,58].

The mechanisms underlying cherubism were elucidated by analyzing SH3BP2 cherubism mutant mice, in which the P416R mutation (equivalent to the most common P418R mutation in human patients) was introduced in the murine *Sh3bp2* locus [52]. Analysis of the SH3BP2 cherubism mutant mice revealed that heterozygous mice exhibit osteopenia owing to increased osteoclast formation, whereas homozygous mutant mice spontaneously develop systemic organ inflammation and severe osteopenia [52]. The mutant macrophages are hypersensitive to receptor activator NF-κB ligand (RANKL) and tumor necrosis factor (TNF), leading to the production of a large number of osteoclasts [52,57,65,66,67]. In the mutant cells, accumulated SH3BP2 proteins enhance phosphorylation of SYK in response to RANKL and TNF, and subsequently activate NFATc1, which is a master regulator of osteoclastogenesis [52,57]. Additionally, the mutant macrophages of homozygous mice are hyperactivated with increased TNF production in response to macrophage colony-stimulating factor (M-CSF) and Toll-like receptor ligands [52,68,69,70], which are also mediated by the accumulated SH3BP2 protein. The highly activated osteoclasts and macrophages are assumed to be involved in the pathogenesis of bone destructive phenotypes associated with cherubism [71].

### 2.2. Increased Osteoclastogenesis Induced by Tankyrase Inhibitors

Poly(ADP-ribose) is expressed in bone tissue, especially in the nucleus and cytoplasm of bone cells [72], suggesting the potential implication of poly(ADP-ribose) and PARPs on bone homeostasis. We recently determined the role of tankyrase in bone cells. Inhibition of tankyrase activity modulates bone homeostasis via the increased stability of SH3BP2 [73]. Although SH3BP2 is a tankyrase substrate [5,6], the research on tankyrase biology to date has not focused on the role of SH3BP2.

Because we previously reported the importance of SH3BP2 for bone homeostasis [52,57], we examined the effects of tankyrase inhibitors on bone metabolism, with a particular focus on SH3BP2. When primary murine bone marrow-derived macrophages were treated with tankyrase inhibitors (XAV939, IWR-1, and G007-LK) in the presence of RANKL, the cells formed a large number of osteoclasts, represented by tartrate-resistant acid phosphatase (TRAP)-positive multi-nucleated giant cells [73]. Additionally, SH3BP2 accumulated in tankyrase inhibitor-treated cells, after which the phosphorylation of SYK and the nuclear translocation of NFATc1 significantly increased, relative to the corresponding levels in untreated cells (Figure 4) [73]. These findings are consistent with the mechanisms underlying the increased osteoclastogenesis observed in SH3BP2 cherubism mutant macrophages [52,57,65]. Moreover, tankyrase inhibitors induced osteoclast differentiation in human peripheral mononuclear cell (PBMC) cultures as well as in murine macrophage cultures [73].

### 2.3. Increased Osteoblast Maturation Induced by Tankyrase Inhibitors

Tankyrase inhibitors also affect osteoblast differentiation and maturation. Tankyrase inhibitor treatments can suppress Wnt signaling, as indicated by increased AXIN contents and the subsequent suppression of the nuclear translocation of β-catenin [11]. We revealed that tankyrase inhibitors can promote osteoblastic mineralization despite their Wnt inhibitory effect, as indicated by the increased mineral deposition [73]. These findings suggest that other molecules promote osteoblastogenesis, despite the suppression of Wnt/β-catenin signaling. This can be explained, at least in part, by the SH3BP2 function in osteoblasts. Specifically, SH3BP2 is produced in osteoblasts, and SH3BP2-deficient mice are reported to exhibit osteopenia due to the dysregulation of osteoblast differentiation and maturation [59]. Additionally, SH3BP2 activates the tyrosine kinase ABL, which is required for the differentiation of osteoblasts along with the transcriptional coactivator TAZ [60]. Indeed, we observed that tankyrase inhibitors increase the abundance of SH3BP2 and the nuclear expression of ABL, TAZ, and RUNX2 (Figure 5) [73]. This suggests that accumulated SH3BP2 contributes to the activation of the ABL–TAZ complex, and consequently to the acceleration of osteoblast differentiation and maturation despite the Wnt-suppressive effect of tankyrase inhibitors.

### 2.4. In Vivo Bone Loss Due to Tankyrase Inhibitors

Bone mass is maintained by balancing bone formation and resorption, which is mediated by osteoblasts and osteoclasts, respectively. To evaluate the in vivo effect of tankyrase inhibition, seven-week-old wild-type mice were orally administered G007-LK for four weeks [73]. This treatment regimen induced tibial and vertebral bone loss [73]. The in vivo administration of G007-LK increased osteoclast formation, which was indicated by an increase in the number of TRAP-positive cells in bone tissues and by the increased abundance of a bone resorption marker in serum, with no changes to the serum bone formation marker contents [73]. These findings suggest that tankyrase inhibitors predominantly affect osteoclasts, but not osteoblasts in vivo. This may be explained by the fact that macrophages/osteoclasts are more susceptible to the inhibitors at low concentrations. Indeed, although tankyrase inhibitors enhance osteoblast differentiation in vitro, the minimum concentration for augmenting osteoblastogenesis is approximately 10 times higher than the concentration for increasing osteoclastogenesis [73]. Furthermore, osteoblastic bone formation may be suppressed by osteoclasts or other cells in vivo via the indirect effects of tankyrase inhibitors. The osteopenic phenotype observed after a tankyrase inhibitor treatment is consistent with the results of an analysis of radiation-induced chimeric mice lacking tankyrase-1 and tankyrase-2 in hematopoietic cells, in which tankyrase activity is conserved in osteoblasts [6].

In our in vivo experiment [73], the G007-LK dose was similar to the dose previously applied to a xenograft model of colorectal cancer [39]. This dose inhibited tumor growth in the xenograft model [39], while a similar dose induced bone loss as described above. These findings suggest that tankyrase inhibitors may exhibit both anti-cancer and osteopenic effects at similar serum concentrations in vivo (Figure 6). If tankyrase inhibitors are administered to humans in a clinical setting, we will likely need to protect against osteoporosis by prescribing anti-bone resorbing agents (e.g., bisphosphonates or an anti-RANKL antibody [74,75]). Additionally, the development of a drug delivery system specifically targeting cancer cells will be required.

### 2.5. Effects of PARP1 Inhibition on Bone

Apart from the roles of tankyrase in bone, it should be noted that there is some evidence that another member of the PARP family, PARP1, also affects bone homeostasis. In terms of osteoclastogenesis, PARP1 is reported to be a negative regulator. PARP1 suppresses Tcirg1 gene expression, which encodes the a3 isoform of the V-ATPase subunit, in murine pre-osteoclastic RAW264.7 cells [76]. Additionally, PARP1 binding to the regulatory region of the Tcirg1 gene is disrupted after RANKL treatment [76]. Reflecting these findings, PARP1 inhibition is reported to enhance osteoclastogenesis. PARP1 silencing or pharmacological inhibition of its enzymatic activity enhances osteoclast differentiation and function in RAW264.7 cells in culture [77]. Furthermore, PARP1-deficient mice exhibit bone loss associated with increased osteoclasts [77]. Mechanistically, PARP1 deficiency promotes osteoclast development via increased IL-1β expression and subsequent Nfatc1/A expression, a master regulator of osteoclastogenesis [77]. Interestingly, the increased osteoclastogenesis is associated with activation of NOD-like receptor family, pyrin domain containing 3 (NLRP3) inflammasome [78,79]. Activation of NLRP3 inflammasome promotes cleavage and subsequent degradation of PARP1 as well as maturation of IL-1β [78,79]. These mechanisms indicate the possible involvement of PARP1 in inflammasome-associated osteolysis observed in rheumatoid arthritis, gout, and psoriatic arthritis [80].

In terms of osteoblasts, PARP1 has a promoting effect on osteoblast differentiation [81]. During osteoblast differentiation, the cells release hydrogen peroxide, which then activates PARP1. Next, the activated PARP1 promotes osteoblast differentiation via p38 MAP kinase-mediated pathways [82,83]. In contrast to the promoting effect of PARP1 on osteoblast differentiation in physiological conditions, PARP1 might suppress bone formation in inflammatory conditions. In inflammatory conditions, TNF inhibits osteoblast differentiation and bone formation [84,85]. A likely mechanism is decreased expression of the PHEX gene, which encodes a zinc endopeptidase expressed in osteoblasts and contributes to bone mineralization [86]. PARP1 is reported to suppress PHEX gene expression cooperatively with NF-κB signaling downstream of TNF receptors in murine cells [86]. Because the effects of PARP1 vary depending on the environment, the role of PARP1 in osteoblasts needs to be clarified in various pathological conditions.

Regarding the effect of PARP1 on bone, SH3BP2 might act as a modifier of PARP1-mediated bone homeostasis, not just tankyrase-mediated bone homeostasis. PARP1 is reported to bind the human *SH3BP2* promoter and regulates SH3BP2 gene expression [87]. In PARP1-deficient murine bone marrow-derived macrophages, *Sh3bp2* promoter activity and SH3BP2 protein expression are significantly suppressed [87]. Though further research will be required to determine whether SH3BP2 is involved in PARP1-mediated bone homeostasis, SH3BP2 might be involved in osteoclast differentiation through two different mechanisms, transcriptional regulation by PARP1 and post-translational regulation by tankyrase.

## 3. Other Possible Effects of Tankyrase Inhibition and Concluding Remarks

Other than the effect of tankyrase inhibitors on bone, we might need to also pay attention to their effect on immune cells. As described in the previous section, SH3BP2 is ubiquitously expressed in various immune cells, such as macrophages and lymphocytes. Previous studies revealed that excessive amounts of SH3BP2 in macrophages enhance the production of inflammatory cytokines [52,65,68] and increase phagocytic activities [88,89]. Additionally, a deficiency in SH3BP2 impairs B-cell functions [90,91]. Because SH3BP2 gain-of-function mutant mice exhibit macrophage-mediated inflammatory phenotypes associated with increased TNF production [52,68], we assessed whether tankyrase inhibitors can induce inflammatory phenotypes. Our previous in vitro RAW264.7 cell culture experiment revealed that tankyrase inhibition does not enhance inflammatory responses in the presence of lipopolysaccharides [73]. Furthermore, the administration of tankyrase inhibitors to mice failed to induce any detectable inflammation [73]. Although we did not observe any activated inflammation in the tankyrase inhibitor-treated macrophages and mice, we should consider the possibility that higher inhibitor concentrations may lead to macrophage-mediated systemic inflammation. Indeed, we previously reported that in murine arthritis models, SH3BP2 gain-of-function mutant mice exhibit more severe inflammation and bone destruction compared with wild-type mice [57,65,66]. These findings suggest that pre-existing inflammatory conditions (e.g., arthritis and chronic infection) of individuals may be aggravated by tankyrase inhibitors.

Tankyrase inhibitors may be toxic to the intestinal tract. When a tankyrase inhibitor (G-631) was administered in a murine xenograft colorectal cancer model, the inhibitor exhibited an anti-cancer effect, but it simultaneously induced severe intestinal toxicity, characterized by multifocal-regionally extensive necrotizing and ulcerative enteritis [92]. Wnt signaling is critical for intestinal tissue homeostasis. Therefore, the intestinal toxicity was likely mediated through the Wnt-suppressive activity of G-631. Because no obvious enteritis was observed in our in vivo administration of G007-LK, an additional investigation may be required to determine if G-631-induced intestinal toxicity is due to tankyrase-specific suppression or if it is a non-specific effect of the drug.

Tankyrase inhibitors are widely recognized as Wnt-specific/selective inhibitors because of their regulatory effects on AXIN [11]. However, as we have described in this article, tankyrase inhibition may modulate various signaling pathways by regulating a variety of substrate proteins, such as SH3BP2 [5,6,73]. Therefore, we should not consider tankyrase inhibitors as Wnt-specific inhibitors, and we must carefully interpret the data when tankyrase inhibitors are used in experiments.

In conclusion, the characterization of tankyrase-mediated cellular processes, especially the AXIN-regulated pathway, has resulted in advances in preclinical research on the utility of tankyrase inhibitors for treating cancers and fibrotic diseases. The next step will involve a more in-depth examination of the relevant cellular processes regulated by SH3BP2 and other substrates to enable researchers to predict and prevent avoidable adverse events associated with tankyrase inhibitors as well as to discover other potential therapeutic applications via modified tankyrase activity.

## Figures and Tables

**Figure 1 cells-08-00195-f001:**
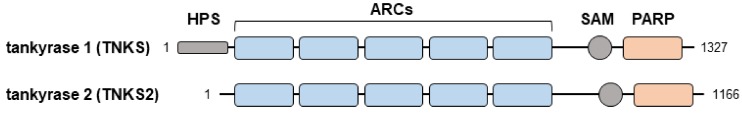
Structure of tankyrase 1 and 2. HPS, homopolymeric runs of histidine, proline, and serine; ARCs, ankyrin repeat clusters; SAM, sterile alpha module; PARP, catalytic domain.

**Figure 2 cells-08-00195-f002:**
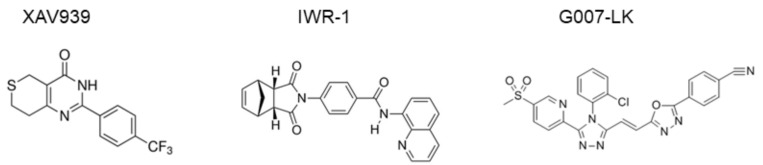
Structure of tankyrase inhibitors showing the structures of XAV939, IWR-1, and G007-LK [50].

**Figure 3 cells-08-00195-f003:**
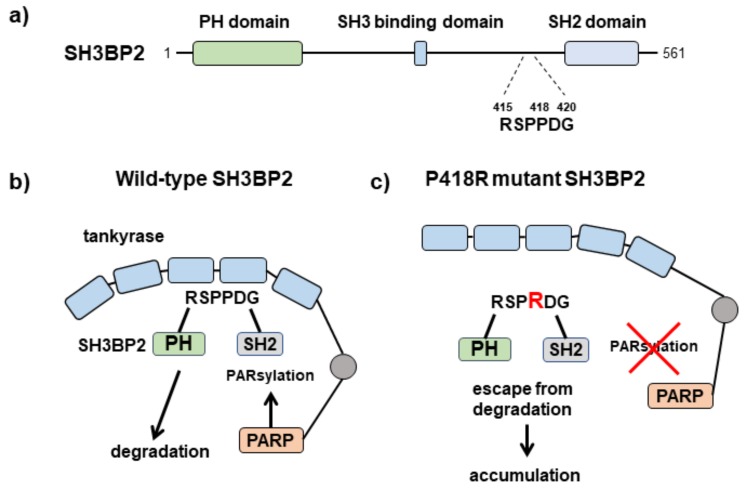
Structure of SH3 domain-binding protein 2 (SH3BP2) and increased SH3BP2 accumulation in SH3BP2 cherubism mutant cells. (**a**) Structure of SH3BP2. SH3BP2 protein consists of three modular domains, namely, pleckstrin homology (PH), SH3-binding, and SH2 domains. Cherubism mutations are located within the RSPPDG sequence. (**b**) Tankyrase recognizes wild-type SH3BP2 protein through the RSPPDG sequence and then induces PARsylation and subsequent degradation of SH3BP2 protein. (**c**) The mutated SH3BP2 sequence disrupts the recognition of tankyrase. The mutant protein escapes from the degradation process, resulting in aberrant accumulation in the cytoplasm.

**Figure 4 cells-08-00195-f004:**
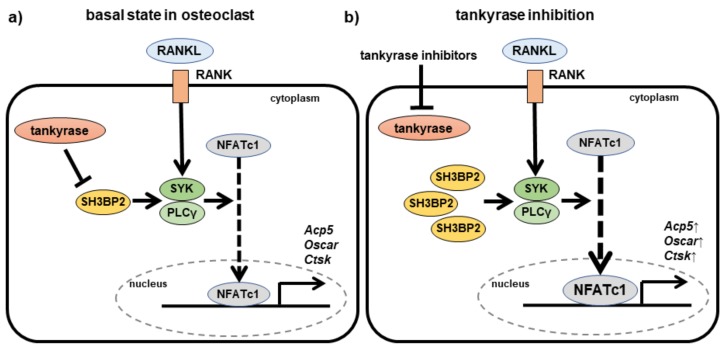
Effects of tankyrase inhibitors in osteoclasts. (**a**) Signaling pathways of osteoclastogenesis in physiological conditions. SH3BP2 transduces osteoclast-inducing signaling pathways as an adaptor protein downstream of RANK. Tankyrase regulates expression of the SH3BP2 protein. (**b**) In the presence of tankyrase inhibitors, the SH3BP2 protein accumulates in the cells in the absence of the tankyrase-mediated degradation process. Accumulated SH3BP2 enhances activation of SYK and PLCγ downstream of RANK, resulting in enhanced osteoclastogenesis, represented by increased expression of osteoclast-associated genes, such as *Acp5*, *Oscar*, *Ctsk*. A solid arrow indicates direct stimulatory modification, a blunt-ended arrow indicates direct inhibitory modification, and a dashed arrow indicates translocation of the protein. RANKL, receptor activator of NF-κB ligand; RANK, receptor activator of NF-κB; SYK, spleen tyrosine kinase, PLCγ, phospholipase C gamma; NFATc1, nuclear factor of activated T-cells, cytoplasmic 1; Acp5, acid phosphatase 5; Oscar, osteoclast-associated receptor; Ctsk, cathepsin K.

**Figure 5 cells-08-00195-f005:**
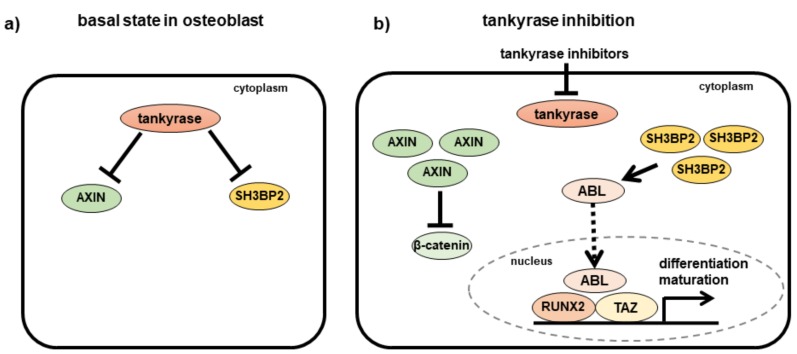
Effects of tankyrase inhibitors in osteoblasts. (**a**) Tankyrase inhibits expression levels of AXIN and SH3BP2 in physiological conditions. (**b**) In the presence of tankyrase inhibitors, both AXIN and SH3BP2 proteins accumulate excessively in the cells. Abundant AXIN suppresses β-catenin signaling. Simultaneously, accumulated SH3BP2 enhances activation of ABL and subsequent RUNX2-mediated gene expression, resulting in increased osteoblast differentiation and maturation. A solid arrow indicates direct stimulatory modification, a blunt-ended arrow indicates direct inhibitory modification, and a dashed arrow indicates translocation of the protein. RUNX2, runt-related transcription factor 2; TAZ, tafazzin.

**Figure 6 cells-08-00195-f006:**
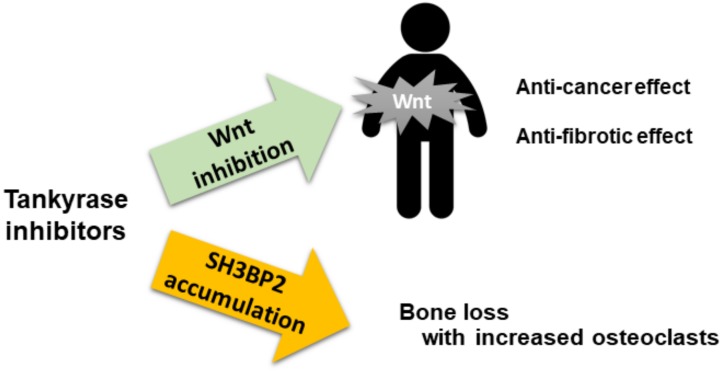
Possible adverse effect of tankyrase inhibitors on bone. Tankyrase inhibitors exert anti-cancer and anti-fibrotic effects via Wnt inhibition. Simultaneously, these inhibitors could induce systemic bone loss with increased number of osteoclasts via SH3BP2 accumulation.

**Table 1 cells-08-00195-t001:** Preclinical studies of tankyrase inhibitors with in vivo murine cancer models.

Disease Models	Cells	Drugs	References
Colorectal cancer xenograft model	COLO-320DM SW430	RK-287107	[16]
Colorectal cancer xenograft model	COLO-320DM HCT-15	G007-LK	[34]
Osteosarcoma xenograft model	MNNG-HOS	IWR-1	[24]
Gastrointestinal stromal tumor model(*Kit^V558Δ/+^* mouse model)	Gastrointestinal stromal tumor	G007-LK	[35]
Orthotopic lung cancer model	HCC4006 H1650 PC9T790M	AZ1366	[36]
Colorectal cancer xenograft model	Colorectal cancer	AZ1366	[37]
Colorectal cancer xenograft model	Colorectal cancer	NVP-TNKS656	[38]
Colorectal cancer xenograft model	COLO-320DM	G007-LK	[39]

**Table 2 cells-08-00195-t002:** Preclinical studies of tankyrase inhibitors with in vivo murine fibrosis models.

Disease Models	Drugs	References
Bleomycin-induced dermal fibrosis model	XAV939	[42]
TGF-β receptor I-overexpressing model	XAV939	[42]
Bleomycin pulmonary fibrosis model	XAV939	[45]

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
