# Peer review of "Tankyrase (PARP5) Inhibition Induces Bone Loss through Accumulation of Its Substrate SH3BP2"

_cells, 2019, doi:10.3390/cells8020195_

Round 1

Reviewer 1 Report

This review by Mukai et al. intends in describing recent findings that tankyrase inhibition induces functional changes in bone cells via the SH3 domain-binding protein 33 (SH3BP2). It is suggested by the authors that such treatment, with tankyrase inhibition, which is promising in cancer treatment, may have adverse effects on bone mass.

However, the title also summarizes the main deficiency of the manuscript: the lack of a clear take-home message which I believe is the main concern raised by this review.

As it is mentioned in the Abstract and the introduction, the main goal of this article is to describe recent findings related to tankyrase inhibitors and bone loss, and this is not clearly indicated in the title.

As a major comment, I recommend to the authors to modify and re-structure the article (and the title) in order to give a clear and well-defined subject.

For instance, the introduction is only a repetition of the abstract without any new information.

The most interesting paragraphs are 5 and 6 that clearly represent the body of the paper and should be organized in such a way to give a clear understanding of the main consequences of the proposed treatment modality.

Even if the article is generally well written, the organization of the paragraphs needs to be modified, the actual introduction paragraph can be removed (or at least modified) since no new information is given and the other paragraph should be replaced as following:

1 - The introduction of the article should aims in presenting:

- PARP superfamily and more specifically tankyrase

- SH3BP2 domain and cherubism disease

- tankyrase inhibitors (anti-cancer, etc…)

2 – Effects of tankyrase inhibitors on bones

This should be the central part of the article

- Increased osteoclastogenesis

- Increased osteoblast maturation

- In vivo bone loss

- Comparison with Parp1 inhibition on bone

3 – Other possible effects of tankyrase inhibitors and Conclusion

Figures

Figures 1, 2 and 3 are highly schematic, and not easy to follow.

Figure 1 A is not informative, and the text is enough.

Figure 1 B: is it an experiment from the authors (protocol?), or taken from the literature? (ref?)

Figure 2: explain the legend: colors, lines, dotted lines, etc…

Figure 3 A: “tankyrase regulates expression levels…” the symbol within the figure schematic is: “inhibits”

Figure 3 B: explain the legend: colors, lines, dotted lines, etc…

Author Response

Response to Reviewers’ Comments

We thank the reviewers for their comments on our manuscript, cells-425440, and for the invitation to submit the revised manuscript. We have carefully considered the comments offered by the four reviewers. In revising the manuscript, we have made substantial changes in response to all issues raised in the initial review. A marked copy of the revised manuscript is uploaded as a “Tankyrase review Cells (Marked copy)” with the changes addressing the reviewers’ comments shown in red for ease of identification by the reviewers. The revisions made in response to the comments of each reviewer are as follows:

Reviewer#1

As a major comment, I recommend to the authors to modify and re-structure the article (and the title) in order to give a clear and well-defined subject.

For instance, the introduction is only a repetition of the abstract without any new information.

The most interesting paragraphs are 5 and 6 that clearly represent the body of the paper and should be organized in such a way to give a clear understanding of the main consequences of the proposed treatment modality.

Even if the article is generally well written, the organization of the paragraphs needs to be modified, the actual introduction paragraph can be removed (or at least modified) since no new information is given and the other paragraph should be replaced as following:

1 - The introduction of the article should aims in presenting:

- PARP superfamily and more specifically tankyrase

- SH3BP2 domain and cherubism disease

- tankyrase inhibitors (anti-cancer, etc…)

2 – Effects of tankyrase inhibitors on bones

This should be the central part of the article

- Increased osteoclastogenesis

- Increased osteoblast maturation

- In vivo bone loss

- Comparison with Parp1 inhibition on bone

3 – Other possible effects of tankyrase inhibitors and Conclusion

Response: We appreciate the reviewer’s constructive and concrete suggestions. We have changed the title to “Tankyrase (PARP5) inhibition induces bone loss through accumulation of its substrate SH3BP2”. We hope this title clearly shows the main subject of this manuscript, as the reviewer suggested.

In terms of the organization of the manuscript, we have deleted the description written in “1. Introduction” in the original manuscript. Also, we have re-organized the paragraphs following the reviewer’s suggestion. The section on SH3BP2 and cherubism has been included in section 2 because it is closely related to the subsequent paragraphs. The section on SH3BP2 has been renamed “SH3BP2, an unappreciated substrate for tankyrase, regulates osteoclastogenesis”. We believe this reorganization would clarify the subject of this manuscript and improve readability, thanks to the reviewer.   

Figures

Figures 1, 2 and 3 are highly schematic, and not easy to follow.

Figure 1 A is not informative, and the text is enough.

Figure 1 B: is it an experiment from the authors (protocol?), or taken from the literature? (ref?)

Figure 2: explain the legend: colors, lines, dotted lines, etc…

Figure 3 A: “tankyrase regulates expression levels…” the symbol within the figure schematic is: “inhibits”

Figure 3 B: explain the legend: colors, lines, dotted lines, etc…

Response: To improve the figures, we have revised them as follows: For Figure 1 (current Figure 3), we have deleted some text and instead added a description in the figure legend. Figure 1B in the original manuscript presented the results of immunoblotting, which we have previously reported (Mukai et al. J Bone Min Res. 2014, 29, 2618-2635). Since Figure 3C and the immunoblotting image are redundant, we have deleted the immunoblotting image in the revised manuscript.

For Figures 2 and 3 (current Figures 4 and 5), we have added a description explaining the meaning of lines and dotted lines as follows: “A solid arrow indicates direct stimulatory modification, a blunt-ended arrow indicates direct inhibitory modification, and a dashed arrow indicates translocation of the protein.”

In terms of the colors, we have colored each protein to improve the visibility, not depending on the functions of the protein. If this style of coloring is not appropriate, we could use monochrome figures if necessary.

Also, we have changed the description “tankyrase regulates expression …” to “tankyrase inhibits expression …” in the figure legend for Figure 3 (current Figure 5), to clarify the meaning as the reviewer suggested.

Reviewer 2 Report

In this review the authors describe the role of Tankyrase in regulating bone homeostasis via its lesser-studied substrate SH3BP2 and the potential of targeting Tankyrase for the treatment of cherubism. Overall, I enjoyed reading this timely review, however, there are some limitations. Please see my comments below to address these issues

1) The authors should include a section where they discuss the concept of chemical inhibition of proteins (e.g., PARP inhibitors) and the success of this combination therapy approach. Useful reference for this concept include

- Bhattacharjee, S. and S. Nandi, DNA damage response and cancer therapeutics through the lens of the Fanconi Anemia DNA repair pathway. Cell Communication and Signaling, 2017. 15.

- Bhattacharjee, S. and S. Nandi, Rare Genetic Diseases with Defects in DNA Repair: Opportunities and Challenges in Orphan Drug Development for Targeted Cancer Therapy. Cancers, 2018. 10(9): p. 298.

2) The figures are poorly made and need improvement. Additionally, there are basic formatting errors that need to be fixed. 

3) Include details of preclinical studies targeting Tankyrase in tabular form. 

4) The manuscript needs to be edited for clarity and grammar. 

Minor points

1) Several references are missing, e.g., Page 2, line 55

Author Response

Response to Reviewers’ Comments

We thank the reviewers for their comments on our manuscript, cells-425440, and for the invitation to submit the revised manuscript. We have carefully considered the comments offered by the four reviewers. In revising the manuscript, we have made substantial changes in response to all issues raised in the initial review. A marked copy of the revised manuscript is uploaded as a “Tankyrase review Cells (Marked copy)” with the changes addressing the reviewers’ comments shown in red for ease of identification by the reviewers. The revisions made in response to the comments of each reviewer are as follows:

Reviewer#2

1) The authors should include a section where they discuss the concept of chemical inhibition of proteins (e.g., PARP inhibitors) and the success of this combination therapy approach. Useful reference for this concept include

- Bhattacharjee, S. and S. Nandi, DNA damage response and cancer therapeutics through the lens of the Fanconi Anemia DNA repair pathway. Cell Communication and Signaling, 2017. 15.

- Bhattacharjee, S. and S. Nandi, Rare Genetic Diseases with Defects in DNA Repair: Opportunities and Challenges in Orphan Drug Development for Targeted Cancer Therapy. Cancers, 2018. 10(9): p. 298.

Response: We have added the following description in section 1.2. on page 2, citing the suggested literature, as follows:

“In addition, the mechanism of synthetic lethality should be noted. Cancers with mutation of BRCA1/2 are treatable by PARP inhibitors [29-31]. PARPs and BRCA1/2 are both important for DNA repair [32,33]. Neither mutation of BRCA nor inhibition of PARP is lethal because cells can inherently repair DNA damage [32,33]. However, inhibiting both pathways, represented by administration of PARP inhibitors in BRCA1/2-mutated cancers, prevents both DNA repair machinery, resulting in unrepairable DNA damage and subsequent cell death of the mutated cancer cells [32,33]. The PARP1 inhibitor olaparib has been approved and has achieved success in the treatment of BRCA-mutated cancers in the basis of this synthetic lethality approach [29-31].”

2) The figures are poorly made and need improvement. Additionally, there are basic formatting errors that need to be fixed.

Response: We have substantially revised the figures to make them easy to understand. We hope the revision improves the formatting and clarity of the figures.

3) Include details of preclinical studies targeting Tankyrase in tabular form.

Response: We have added 2 tables on pages 2 and 3 to summarize preclinical studies. Table 1 is for murine in vivo cancer models, and Table 2 is for murine in vivo fibrosis models.

4) The manuscript needs to be edited for clarity and grammar.

Response: Our manuscript has been edited twice by an English language editing service ThinkSCIENCE Inc. (https://thinkscience.co.jp/en/). We hope the manuscript is now satisfactorily edited. The certificate of English editing is attached.

Minor points

1)    Several references are missing, e.g., Page 2, line 55

Response: We have added the corresponding references.

Reviewer 3 Report

The manuscript titled ‘Tankyrase (PARP5) regulates bone homeostasis via its substrate SH3BP2’ reviews literature describing Tankyrase and its inhibition effects on bone homeostasis. The manuscript is written and organized well. It could still be refined for better impact.

1)    Structures of Tankyrase inhibitors should be provided. 

2)    Structures of Tankyrase and SH3BP2 proteins could be provided for graphical presentation.

3)    Re-check the references: Ref #11 and #39 are same.

4)    Figures 2 and 3 are repetitive to some extent. These could be merged into one figure. Also, a general introduction/conclusion figure could be provided which summarized the overall effects of Tankyrase and its inhibition.

Author Response

Response to Reviewers’ Comments

We thank the reviewers for their comments on our manuscript, cells-425440, and for the invitation to submit the revised manuscript. We have carefully considered the comments offered by the four reviewers. In revising the manuscript, we have made substantial changes in response to all issues raised in the initial review. A marked copy of the revised manuscript is uploaded as a “Tankyrase review Cells (Marked copy)” with the changes addressing the reviewers’ comments shown in red for ease of identification by the reviewers. The revisions made in response to the comments of each reviewer are as follows:

Reviewer#3

1) Structures of Tankyrase inhibitors should be provided.

Response: We have added Figure 2, which presents the structures of XAV939, IWR-1, and G007-LK. We selected these three drugs because they were used in our previous study (Fujita et al. Bone 2018, 106, 156-166) and are referenced in the subsequent Section 2. 

2Structures of Tankyrase and SH3BP2 proteins could be provided for graphical presentation.

Response: We appreciate the reviewer’s suggestion. We have prepared the figures to present the structures of tankyrase (Figure 1) and SH3BP2 (Figure 3), and then revised Figure 3B to clarify the interaction between tankyrase and SH3BP2.

3Re-check the references: Ref #11 and #39 are same.

Response: Thank you for pointing this out. We have corrected the referencing accordingly.

4-1Figures 2 and 3 are repetitive to some extent. These could be merged into one figure.

Response: We think this comment is raised because of our inadequate description. We tried to describe the schematic mechanisms of osteoclasts in Figure 2 (current Figure 4) and those of osteoblasts in Figure 3 (current Figure 5). We have added these descriptions in the figures and legends to avoid misunderstanding. We hope the revision would improve readability.

4-2) Also, a general introduction/conclusion figure could be provided which summarized the overall effects of Tankyrase and its inhibition.

Response: We have added Figure 6, which briefly summarizes the effect of tankyrase inhibition. We hope this figure would clarify the main subject of this manuscript.

Reviewer 4 Report

The paper by Mukai et al. describes interesting results about the involvement of PARP5 in  bone homeostasis and underlines that its inhibitors can be useful in clinical setting as they seem to be associated with adverse effects on bone mass.

The presentation is synthetic overall the review and this is in favour of clarity, by considering the various cellular activities involved. Most of the text is focused on the value of PARP5 inhibitors, in particular those object of the studies by the Authors.

It is satisfactory mainly for a reader not specifically expert in PARP 5 research.

References are updated.

We recommend publication once solving the following points.

Major points

-Abstract :the sentence lines 10-12, might be better at line 13, before “in this review..”.

-In “Instructions for Authors of Cells”, it is recommended that  “..The introduction should briefly place the study in a broad context and highlight why it is important. It should define the purpose of the work and its significance, including specific hypotheses being tested. The current state of the research field should be reviewed carefully and key publications cited.”

Actually in the paper by Mukai et al. Introduction is very short and does not fit the above recommendations. Moreover no references are included.

We suggest to integrate this part in §2, and considering the whole as” Introduction”.

Minor points:

Lines 26-7 “ An increasing number of studies have revealed the various anti-cancer effects of tankyrase inhibitors. References??

l. 74 - an anti-cancer effects>an … effect

A reading through the text  is suggested to correct typing/English mistakes.

It is satisfactory mainly for a reader not specifically expert in PARP 5 research.

References are updated.

We recommend publication once solving the following points.

Author Response

Response to Reviewers’ Comments

We thank the reviewers for their comments on our manuscript, cells-425440, and for the invitation to submit the revised manuscript. We have carefully considered the comments offered by the four reviewers. In revising the manuscript, we have made substantial changes in response to all issues raised in the initial review. A marked copy of the revised manuscript is uploaded as a “Tankyrase review Cells (Marked copy)” with the changes addressing the reviewers’ comments shown in red for ease of identification by the reviewers. The revisions made in response to the comments of each reviewer are as follows:

Reviewer #4

Major points

-Abstract :the sentence lines 10-12, might be better at line 13, before “in this review..”.

Response: We have changed the description in line with the reviewer’s suggestion.

-In “Instructions for Authors of Cells”, it is recommended that  “..The introduction should briefly place the study in a broad context and highlight why it is important. It should define the purpose of the work and its significance, including specific hypotheses being tested. The current state of the research field should be reviewed carefully and key publications cited.”

Actually in the paper by Mukai et al. Introduction is very short and does not fit the above recommendations. Moreover no references are included.

We suggest to integrate this part in §2, and considering the whole as” Introduction”.

Response: As reviewer #1 suggested, we have deleted the “1. Introduction” in the original manuscript to avoid redundancy with the description in the Abstract. Also, we have reorganized the structure of the Introduction, which in the revised manuscript now consists of the subheadings “Overview of tankyrase”, “Preclinical application of tankyrase inhibitors in cancer”, “Preclinical application of tankyrase inhibitors in fibrotic diseases”, and “Tankyrase-specific inhibitors”. We hope this reorganization will resolve the reviewer’s concern.

Minor points:

Lines 26-7 “ An increasing number of studies have revealed the various anti-cancer effects of tankyrase inhibitors. References??

Response: Thank you for pointing this out. In the revised manuscript, that section has been deleted for the reason described above. Instead, references to support evidence of anti-cancer effects are cited in section 1.2. of the revised manuscript with more detailed description.

l. 74 - an anti-cancer effects>an … effect

Response: We have corrected this error.

A reading through the text  is suggested to correct typing/English mistak

Response: Our manuscript has been edited twice by an English language editing service ThinkSCIENCE Inc. (https://thinkscience.co.jp/en/). We hope the manuscript is now satisfactorily edited. The certificate of English language editing is attached.

Round 2

Reviewer 1 Report

I accept the corrections

Reviewer 2 Report

The authors have satisfactorily responded to my comments.

Reviewer 4 Report

The manuscript is much improved compared to the previous version.

I have no more suggestions and recommend its publication.